

# Statistical physics of the polarised IKKT matrix model

Sean A. Hartnoll and Jun Liu

Department of Applied Mathematics and Theoretical Physics,
University of Cambridge, Cambridge CB3 0WA, UK

## Abstract

The polarised IKKT matrix model is the worldpoint theory of $N$ D-instantons in a background three-form flux of magnitude $\Omega$, and promises to be a highly tractable model of holography. The matrix integral can be viewed as a statistical physics partition function with inverse temperature $\Omega^4$. At large $\Omega$ the model is dominated by a matrix configuration corresponding to a 'polarised' spherical D1-brane. We show that at a critical value of $\Omega^2 N$ the model undergoes a first order phase transition, corresponding to tunneling into a collection of well-separated D-instantons. These instantons are the remnant of a competing saddle in the high $\Omega$ phase corresponding to spherical $(p,q)$ fivebranes. We use a combination of numerical and analytical arguments to capture the different regimes of the model.


# 1 Introduction

Recent works [1–3] have revived a supersymmetric mass deformation [4] of the IKKT matrix model [5], and demonstrated that it constitutes a viable model of holographic duality and emergent spacetime. The IKKT model is the worldpoint theory of $N$ D-instantons in type IIB string theory, and the mass deformation captures the polarisation of the D-instantons into higher dimensional Euclidean branes by background fluxes, in the spirit of the Myers effect [6]. The model has the form of a statistical physics partition function, with neither time nor space. The absence of a quantum mechanical time is the source of some conceptual challenges, but means that the model is highly tractable. In particular, supersymmetric localisation can be used to express the partition function exactly as a sum over 'fuzzy sphere' matrix configurations, labeled by $N$ dimensional representations $\mathcal{R}$ of $\mathfrak{su}(2)$, such that [3]

$$Z = \sum_{\mathcal{R}} Z_{\mathcal{R}}, \qquad Z_{\mathcal{R}} = C_{\mathcal{R}} e^{-S_{\mathcal{R}}^0} \int d\vec{m}_{\mathcal{R}} \, e^{-S_{\mathcal{R}}^{\text{eff}}(\vec{m}_{\mathcal{R}})}. \qquad (1)$$

We will shortly give formulae for all quantities appearing here. For each representation there are integrals over moduli $\vec{m}_{\mathcal{R}}$, one modulus for each irreducible representation appearing in $\mathcal{R}$. In a localisation calculation, fluctuation effects about the supersymmetric fuzzy sphere minima are packaged exactly into the normalisation factors $C_{\mathcal{R}}$ and moduli space measure $e^{-S_{\mathcal{R}}^{\text{eff}}}$.

The expression (1) is the starting point of this paper. We will not write down the polarised IKKT matrix action explicitly, see [1–3]. Our objective is to understand the statistical physics of (1), and its connection to emergent spacetime physics. The 'energetics' of the partition function is determined by the action $S_{\mathcal{R}}^0$ of the fuzzy sphere configurations

$$S_{\mathcal{R}}^0 = -\frac{9\Omega^4}{2^{15}} \sum_M n_M M(M^2 - 1). \qquad (2)$$

Here $\Omega$ is the mass deformation parameter. The partition function $Z$ is a function of $\Omega$ and $N$ only. We may think of the polarised IKKT model as a canonical partition function with $\Omega^4$ being the inverse temperature. In (2) the irreducible representation of dimension $M$ appears with degeneracy $n_M$ in the decomposition of $\mathcal{R}$. That is, the sum over representations in (1) is equivalent to a sum over partitions $\{n_M\}$ such that

$$\sum_M n_M M = N. \qquad (3)$$

In (2) we see that the supersymmetric fuzzy spheres do not all have the same action. This is distinct from, for example, the mass-deformed BMN matrix quantum mechanics in which all representations have the same energy [7]. In particular, in the large $\Omega \to \infty$ limit the partition function is dominated by the maximally irreducible fuzzy sphere. This has $n_N = 1$ and hence the lowest action (2). This is the 'low temperature' limit of the partition function, in which energy strongly dominates over entropy. The matrix partition function can be evaluated by saddle point in this limit [1], without using localisation. For the $U(N)$ theory at large $\Omega$

$$\left. \frac{\log Z_{\text{irred}}}{N} \right|_{\Omega \to \infty} \approx c_N + 2\left(\frac{3\,\Omega^2 N}{2^8}\right)^2 + \mathcal{O}(1/N). \qquad (4)$$

We are primarily interested in the thermodynamic large $N$ limit of the partition function. The constant $c_N \equiv \frac{N}{2}\left(\frac{3}{2} + \log \frac{(2\pi)^{10}}{N}\right)$. At large $\Omega$ the maximally irreducible fuzzy sphere saddle describes a spherical, classical probe D1-brane in a background with nonvanishing NSNS flux

and axion [1]. The D1-brane arises as the polarisation of $N$ D-instantons by the background flux. Fluctuations of the D1-brane geometry are described by a Maxwell field and a scalar.

In the opposite $\Omega \to 0$ limit, the action (2) becomes the same for each representation. 'Entropic' effects are therefore crucial in this high temperature regime. The moduli integrals in (1) are seen to each develop a $1/\Omega^2$ divergence in this limit, and hence the partition function is dominated by the maximally reducible representation which has the most moduli integrals. The constraint (3) allows $N$ copies of the one-dimensional representation, so that $n_1 = N$. The small $\Omega$ divergences allow the partition function to be evaluated in this limit, again without using localisation [3]

$$\left.\frac{\log Z_{\mathrm{triv}}}{N}\right|_{\Omega \to 0} \approx c_N + \frac{1}{2}\log\frac{2e^2}{3\pi} + \log\frac{2^8}{3\Omega^2 N} + \mathcal{O}(1/N). \tag{5}$$

In this limit the dominant configurations should be thought of as $N$ well-separated D-instantons. These are captured by approximately commuting matrices, with the simultaneous eigenvalues giving the location of the D-instantons. We will see in §6 below that at larger values of $\Omega$ the D-instantons in this 'trivial', maximally reducible, representation coalesce to form $(p, q)$ fivebrane bound states of NS5- and D5-branes.

The polarised IKKT model therefore exhibits the familiar behavior of a canonical partition function: a low temperature phase dominated by energy gives way to a high temperature phase dominated by entropy. In the thermodynamic large $N$ limit, the phases may be distinguished by the expectation value of the $\mathfrak{su}(2)$ Casimir. On a given representation the trace of the Casimir is

$$\frac{1}{N}\mathrm{Tr}_{\mathcal{R}}C_2 \equiv \frac{1}{4N}\sum_M n_M M(M^2 - 1). \tag{6}$$

This expression is proportional to the on-shell action (2). At large $\Omega$ the above discussion implies that $\langle\frac{1}{N}\mathrm{Tr}\,C_2\rangle \approx \frac{1}{4}N^2$, while at small $\Omega$ it follows that $\langle\frac{1}{N}\mathrm{Tr}\,C_2\rangle = 0$. It is natural to suspect that there is a phase transition between these two regimes. Comparing the limiting behaviours (4) and (5), one can anticipate that at large $N$ there will be a first order phase transition at

$$\frac{3\Omega^2 N}{2^8} = \mathcal{O}(1). \tag{7}$$

In §2, immediately below, we will find evidence for such a first order transition by performing a numerical evaluation of the full partition function (1). In particular, the Casimir will be found to jump abruptly between the two phases.

Phase transitions out of fuzzy sphere configurations have been seen previously in other matrix models. Such a transition was found by Monte Carlo simulation of a three-matrix integral [8, 9], and an analogous quantum phase transition in the ground state of a three-matrix quantum mechanics was found in [10], using a neural network variational wavefunction. Monte Carlo simulations of the thermal partition function of the string-theoretic BMN matrix quantum mechanics also exhibit the desintegration of a fuzzy sphere saddle, which is usually interpreted as a deconfinement transition [11–14]. A similar mass-induced transition is seen [15] in the Lorentzian IKKT model introduced in [16]. The localisation formula (1) allows us to study the dynamics of a maximally supersymmetric model, with an explicit string-theoretic realisation, without needing to perform intensive matrix Monte Carlo simulations.

The structure of the paper is as follows. In §2 we evaluate the partition function numerically, with the results shown in Figs. 2 and 3. The high $\Omega$ phase shows the featureless dominance of the maximally irreducible representation — which at large $\Omega$ has the spacetime interpretation of a probe D1-brane [1]. In §3 and §4 we show that the low $\Omega$ phase is dominated by a highly reducible representation that is mostly, but not entirely, built from copies of the one-dimensional irrep — at low $\Omega$ such representations have the spacetime interpretation

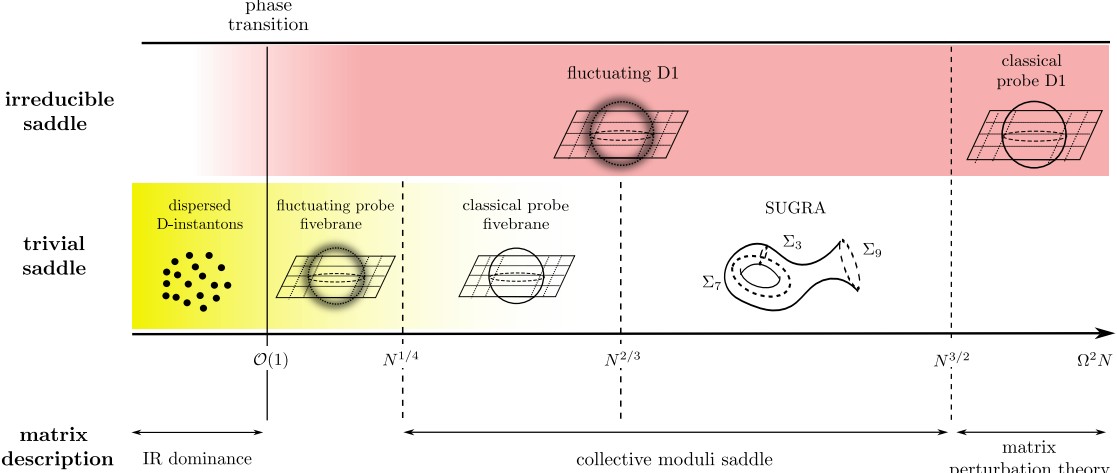

Figure 1: **Phase structure** of the polarised IKKT model. Top row: spacetime physics of the irreducible representation, which is dominant at large $\Omega^2 N$ (shaded red). Middle row: spacetime physics of the maximally reducible or 'trivial' representation, which is dominant at small $\Omega^2 N$ (shaded yellow). Bottom row: appropriate description of the matrix degrees of freedom. The various regimes shown are discussed throughout the paper, and so this figure may be useful as a roadmap. Matrix perturbation theory is discussed in [1].

of a gas of D-instantons. As $\Omega$ is increased through the transition, and the highly reducible representations become sub-dominant, they can eventually be described by a weakly curved supergravity background carrying NS5, D5 and D1 charges [2]. The connection between representations and brane charges is reviewed in §5. In §6 we show that these geometries arise from placing spherical $(p, q)$ fivebranes in a background flux. Towards the transition, however, it is necessary to work in a different duality frame involving a single probe spherical D5-brane. The various regimes are summarised in Fig. 1. We conclude in §7 with an interpretation of the phase transition in terms of spontaneous symmetry breaking and a discussion of the implications of our results for understanding emergent spacetime in the polarised IKKT model.

## 2 A first order phase transition

In this section we evaluate the partition function (1) numerically. We first define the remaining terms in (1). The normalisation factor is [3]

$$C_{\mathcal{R}} = \frac{(2\pi)^{5N^2 + N/2}}{G(N+1)} \prod_M \left[ \frac{1}{n_M!} \left( \frac{32}{3\pi M} \prod_{J=1}^{M-1} \frac{(2+3J)^3}{(1+3J)^3} \right)^{n_M} \right]. \tag{8}$$

Here $G$ is the Barnes $G$-function. The effective action is [3]

$$S_{\mathcal{R}}^{\text{eff}} = \frac{3\,\Omega^4}{2^7} \sum_M \sum_{i=1}^{n_M} M m_{Mi}^2. \tag{9}$$

This action is positive, in contrast to the fuzzy sphere action (2). The moduli measure is

$$d\vec{m}_{\mathcal{R}} = \prod_{M=1}^{N} \prod_{i=1}^{n_M} dm_{Mi}\, Z_{\text{1-loop}}. \tag{10}$$

There is one integral for every irreducible factor of the representation. Recall that the factor with dimension $M$ occurs $n_M$ times. The '1-loop' term [3]

$$Z_{\text{1-loop}} = \prod_{(Mi, Lj)} \frac{f_{\frac{1}{2}|M-L|}(m_{Mi} - m_{Lj})}{f_{\frac{1}{2}|M+L|}(m_{Mi} - m_{Lj})},\tag{11}$$

where the sum is over all pairs of moduli $(Mi, Lj)$ and

$$f_K(x) = \frac{1}{K^2 + \left(\frac{8x}{3}\right)^2} \left| \frac{\Gamma\left(K + \frac{2}{3} + \frac{8ix}{3}\right)}{\Gamma\left(K + \frac{1}{3} + \frac{8ix}{3}\right)} \right|^6.\tag{12}$$

We will evaluate the moduli space integrals using Monte Carlo methods. The dimension of the integral is $D \equiv \sum_M n_M$. We find that the integrals can be performed efficiently by using the effective action as a Gaussian sampling function. That is, we write the moduli integral as

$$\mathcal{I}_{\mathcal{R}} \equiv \int_{\mathbb{R}^D} dm\, z(m) p(m),\tag{13}$$

where $z(m) = Z_{\text{1-loop}}$ and $p(m) = e^{-S_{\mathcal{R}}^{\text{eff}}}$. The sampling function $p(m)$ is a product of Gaussian distributions in the $m_{Mi}$ with widths given by $\frac{2^3}{\sqrt{3M}\Omega^2}$. Generating random samples from this distribution will help to reduce statistical fluctuations of the integration result [17]. Similar methods have been used in [18]. Let us draw $\mathcal{N}$ independent random configurations $\{m_a\}_{a=1}^{\mathcal{N}}$ with probability distribution proportional to $p(m)$. Each $m_a$ is itself a $D$-tuple. The integral (13) is then estimated as

$$\mathcal{I}_{\mathcal{R}} \approx \frac{V_p}{\mathcal{N}} \sum_{a=1}^{\mathcal{N}} z(m_a).\tag{14}$$

Here $V_p = \int_{\mathbb{R}^D} dm\, p(m) = \prod_M \left(\frac{2^3\sqrt{2\pi}}{\sqrt{3M}\Omega^2}\right)^{n_M}$ is the normalisation factor for $p(m)$. The statistical error in the estimate (14) is given by

$$\sigma_{\mathcal{I}_{\mathcal{R}}}^2 = \left(\frac{V_p}{\mathcal{N}}\right)^2 \left[ \sum_{a=1}^{\mathcal{N}} z(m_a)^2 - \frac{1}{\mathcal{N}}\left(\sum_{a=1}^{\mathcal{N}} z(m_a)\right)^2 \right].\tag{15}$$

From (15) one may obtain the statistical error for all quantities defined in terms of the $\mathcal{I}_{\mathcal{R}}$, in the usual way. With $\mathcal{N} = 100$ we find that the error on the observables computed below, which are sums over representations, is less than a few per cent. Some individual $\mathcal{I}_{\mathcal{R}}$ have larger errors but are subdominant in the final results.

With the numerical expression (14) at hand, we obtain the partition function and Casimir as

$$Z = \sum_{\mathcal{R}} Z_{\mathcal{R}} \equiv \sum_{\mathcal{R}} C_{\mathcal{R}} \mathcal{I}_{\mathcal{R}} e^{-S_{\mathcal{R}}^0}, \qquad \left\langle \frac{1}{N}\text{Tr}\, C_2 \right\rangle = \sum_{\mathcal{R}} \frac{Z_{\mathcal{R}}}{Z} \frac{\text{Tr}_{\mathcal{R}} C_2}{N}.\tag{16}$$

These sums are over partitions, the number of which grows rapidly with $N$. We have been able to perform the sums exactly, using a laptop, for $N = 40$. It is likely possible to get to much higher $N$ by statistically sampling from the partitions, but this has not proved necessary for our purposes. Our results for the quantities (16) are shown in Figs. 2 and 3.

Fig. 2 shows $\log Z$ as a function of $\Omega$. The dashed lines show the asymptotic behaviour at large and small $\Omega$ (and large $N$). In these limits the partition function is dominated by the maximally irreducible or the maximally reducible 'trivial' representation, respectively. The partition function is seen to exhibit a kink at a value of $\Omega$ consistent with the estimate (7) for

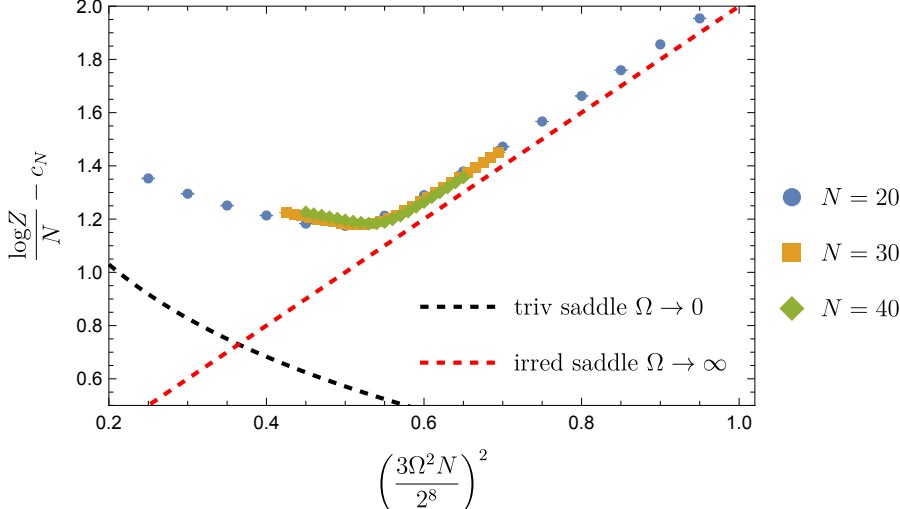

Figure 2: **Logarithm of the partition function** against the deformation parameter $\Omega$. Data points are generated using (16) and $\mathcal{N} = 100$, with the values of $N$ shown in the legend. Dashed lines show the large $N$ expressions (4) and (5) for the large and small $\Omega$ asymptotics, respectively.

the crossover of dominance between these two saddles. On the large $\Omega$ side, the asymptotic behaviour is followed down to the transition. This indicates the continued dominance of a single representation. On the small $\Omega$ side, however, there is a deviation from the asymptotic behaviour. We will see in detail in §3 and §4 below that this is due to a spread in representations contributing.

Fig. 3 shows the expectation value of the Casimir as a function of $\Omega$. This plot shows a first order transition at large $N$ between the maximally irreducible and almost-trivial representations: the high $\Omega$ regime has $\langle \frac{1}{N} \text{Tr} \, C_2 \rangle \approx \frac{1}{4} N^2$, while the low $\Omega$ regime has $\langle \frac{1}{N} \text{Tr} \, C_2 \rangle \approx 0$. Following the midpoint of the transition as a function of increasing $N$ and extrapolating to $N \to \infty$ suggests, see Appendix A for the finite size scaling analysis, that the large $N$ transition occurs at

$$\left( \frac{3 \, \Omega^2 N}{2^8} \right)^2 \approx 0.58 \,. \tag{17}$$

While a first order transition is the most natural interpretation of the numerical results, these cannot exclude some subtle continuity leading to a higher order transition.

## 3 Small deformation and the grand canonical ensemble

Sums over partitions of $N$ can be evaluated more easily in a grand canonical ensemble where the number $N$ is not fixed. The difficulty in the present case is the moduli space integral. However, in the small $\Omega$ limit the moduli integral can be done explicitly for all saddles, leading to [3]

$$Z_{\mathcal{R}} \approx a_N \prod_M \left[ \frac{1}{n_M!} \left( \frac{b_M}{\Omega^2} \right)^{n_M} \right] \tag{18}$$

$$\equiv \frac{(2\pi)^{5N^2 + N/2}}{G(N+1)} \prod_M \left[ \frac{1}{n_M!} \left( \frac{2^8}{(3M)^{3/2} \Omega^2} \sqrt{\frac{2}{\pi}} \prod_{J=1}^{M-1} \frac{(2+3J)^3}{(1+3J)^3} \right)^{n_M} \right] \,. \tag{19}$$

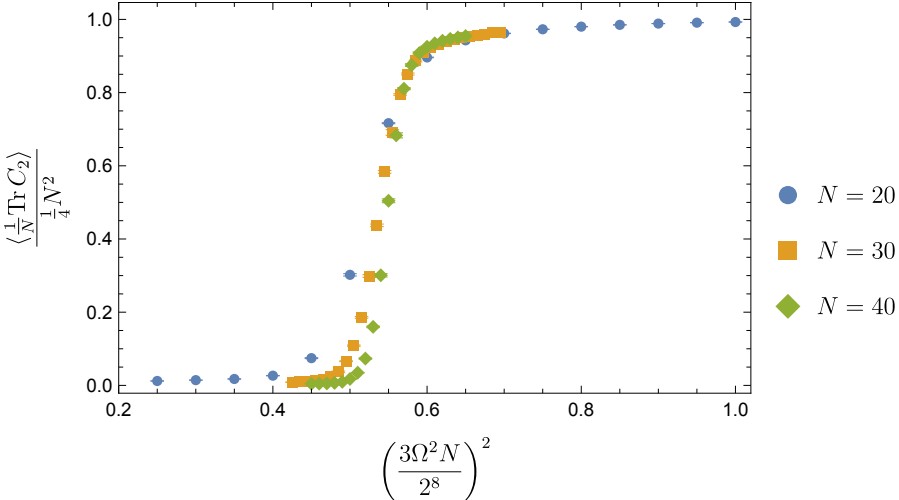

Figure 3: **The $\mathfrak{su}(2)$ Casimir** against the deformation parameter $\Omega$. Data points are as in Fig. 2.

More precisely, it can be verified that when $\Omega^2 N \ll 1$ the '1-loop' part (11) of the moduli integral can be neglected. In this regime the moduli integral becomes $D$ independent Gaussian integrals, giving (19). In this limit we can put the grand canonical ensemble to good use.

The grand canonical computations below are analogous to the statistical description of BMN ground states in [19], and earlier work on half-BPS geometries of SYM theory [20]. Those works aimed to produce a geometry that coarse-grained over individual microstates. The polarised IKKT model has a similar correspondence between representations and space-time geometries [2], see §5 below. However, the matrix configurations that dominate at small $\Omega^2 N$ correspond to well-separated D-instantons. This is consistent with the fact that the approximate partition functions (19) are obtained from independent Gaussian integrals for the moduli, with no collective effects. In this limit the matrices do not holographically generate a weakly curved geometry.

We are thinking of the partition function $Z$ as a canonical partition function, where $\Omega^4$ is the inverse temperature. The grand canonical ensemble additionally has a chemical potential $\beta$ for the 'particle number' $N = \sum_M n_M M$. We will use the grand canonical ensemble to extract the individual expectation values $\langle n_M \rangle$. To this end we also introduce sources $\alpha_M$, via the term $\sum_M \alpha_M n_M$. Using (19), the grand canonical partition function in the small $\Omega^2 N$ limit is

$$\widetilde{Z} = a_N \sum_{\{n_M\}} \prod_M \left[ \frac{1}{n_M!} \left( \frac{b_M}{\Omega^2} \right)^{n_M} e^{(\alpha_M - \beta M) n_M} \right]. \tag{20}$$

Crucially, the $\{n_M\}$ are now unconstrained. We are going to keep the prefactor $a_N$ in terms of $N$ for simplicity. Our objective here is to impose the partition constraint (3) as an expectation value, and for this purpose there is no need to write the prefactor in terms of $\beta$ rather than $N$.

The unconstrained sums in (20) can be performed one by one to obtain

$$\widetilde{Z} = a_N \prod_M \exp\left[ \frac{b_M}{\Omega^2} e^{\alpha_M - \beta M} \right]. \tag{21}$$

The expectation values thus follow an 'isochemical' distribution with a classical Boltzmann factor,

$$\langle n_M \rangle = \frac{\partial \log \widetilde{Z}}{\partial \alpha_M} \bigg|_{\alpha_M = 0} = \frac{b_M}{\Omega^2} e^{-\beta M}. \tag{22}$$

The grand canonical partition function (21) can then be written as

$$\log \widetilde{Z}\big|_{\alpha_M=0} = \log a_N + \sum_M \langle n_M \rangle. \tag{23}$$

To go back to the canonical partition function one must perform a Legendre transform:

$$\log Z = \log \widetilde{Z} - \beta N, \tag{24}$$

where the chemical potential $\beta$ is fixed in terms of $N$ by imposing (3) as an expectation value

$$N = \sum_M M \langle n_M \rangle = \sum_M \frac{b_M M}{\Omega^2} e^{-\beta M}. \tag{25}$$

We are working at small $\Omega^2 N$. In this limit the constraint (25) requires $\beta$ to be large. It follows that, to leading order, only the $M = 1$ term in the sum in (25) contributes and we have

$$\beta \approx \frac{1}{2} \log\left[ \frac{2}{3\pi} \left( \frac{2^8}{3\,\Omega^2 N} \right)^2 \right]. \tag{26}$$

The expectation values (22) are therefore

$$\langle n_M \rangle \approx \frac{N b_M}{b_1} \left( \frac{\Omega^2 N}{b_1} \right)^{M-1} = \langle n_1 \rangle \left( \frac{\Omega^2 N}{b_1} \right)^{M-1}. \tag{27}$$

As expected, the one-dimensional $M = 1$ irreducible representation is the most common building block of the representation at small $\Omega^2 N$. At any nonzero $\Omega^2 N$ the full representation is not entirely trivial, due to the Boltzmann population of nontrivial representations.

To characterise the spread of representations about the trivial representation more precisely, we can look at the mean and variance of the Casimir (6). The variance of the $n_M$ is given by

$$\sigma_{n_M}^2 = \frac{\partial^2 \log \widetilde{Z}}{\partial \alpha_M^2} = \langle n_M \rangle. \tag{28}$$

Here we used the grand canonical partition function (21). The mean and variance of the Casimir are both given in terms of $n_2$, to leading nontrivial order, as the one-dimensional representation doesn't contribute to the Casimir and higher representations are suppressed. That is

$$\left\langle \tfrac{1}{N}\mathrm{Tr}\, C_2 \right\rangle \approx \frac{6}{4N} \langle n_2 \rangle \sim \Omega^2 N, \qquad \sigma_{\frac{1}{N}\mathrm{Tr}\, C_2}^2 \approx \left( \frac{6}{4N} \right)^2 \langle n_2 \rangle \sim \frac{\Omega^2 N}{N}. \tag{29}$$

In particular, $\left\langle \tfrac{1}{N}\mathrm{Tr}\, C_2 \right\rangle \ll \tfrac{1}{4} N^2$ as was seen in Fig. 3. It is instructive to consider the ratio

$$\frac{\sigma_{\frac{1}{N}\mathrm{Tr}\, C_2}^2}{\langle \tfrac{1}{N}\mathrm{Tr}\, C_2 \rangle^2} \sim \frac{1}{\Omega^2 N} \frac{1}{N}. \tag{30}$$

As $\Omega^2 N$ is increased, at fixed $N$, the width of the distribution decreases relative to its average. This predicts that the distribution of Casimirs will increasingly detach from the origin, as the average grows proportionally to $\Omega^2 N$ while the width grows more slowly.

We have verified the grand canonical predictions in Fig. 4, which shows a numerical computation of the entire probability distribution $P(\tfrac{1}{N}\mathrm{Tr}\, C_2)$. The numerics have been done using both the full partition function and also the small deformation approximation (19). The results agree over this range. The probability distribution is obtained in both cases by adding up the probabilities $Z_{\mathcal{R}}/Z$ for all representations with a given value of $\tfrac{1}{N}\mathrm{Tr}\, C_2$. It is important to

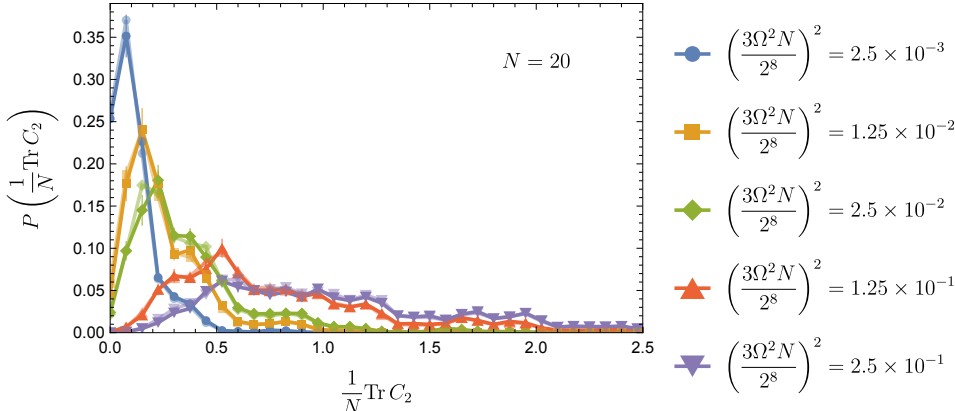

Figure 4: **Probability distribution of the Casimir** for small values of $\Omega^2 N$, calculated with $N = 20$. Each point gives the probability of the Casimir, not the probability density. Solid lines are obtained from the full partition function, while the translucent lines are obtained using the small deformation approximation (19). The two sets of curves are almost indistinguishable.

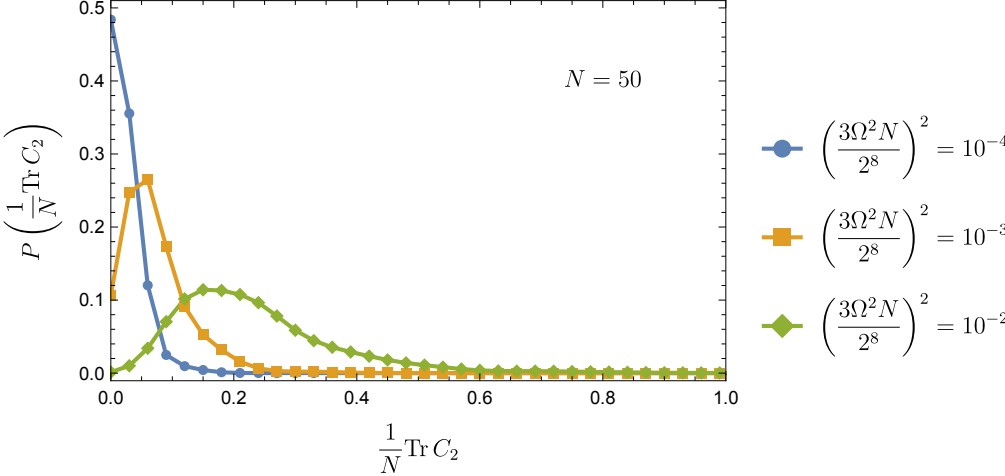

Figure 5: **Probability distribution of the Casimir** for several very small values of $\Omega^2 N$, calculated at $N = 50$ using the small deformation approximation (19).

note that the plot shows the probabilities of each Casimir, not the probability density. That is, the points should sum up to one but the area under the curve will not integrate to one.

Fig. 4 shows that even while the one-dimensional irreducible representation is the most common building block in (27), the distribution of the Casimir is peaked away from zero. This implies that a small but nontrivial representation dominates the partition function for $\Omega^2 N \gtrsim 1/N$. Conversely, as $\Omega^2 N$ is decreased to very small values, the distribution eventually becomes peaked at the origin. We have illustrated this phenomenon in Fig. 5. We used the approximate partition function (19) to make this plot, as it allowed us to work at larger $N$. We have verified that the grand canonical average (29) correctly predicts the location of the peaks in these plots.

The ratio of the variance to the average given in (30) also implies that the Casimir distribution becomes strongly peaked as $N \to \infty$ at fixed $\Omega^2 N$. Indeed, each of the $n_M$ is strongly peaked on its average (22) in this limit. This means that there is a single dominant representation in the partition function at large $N$. While we often refer to this dominant representation

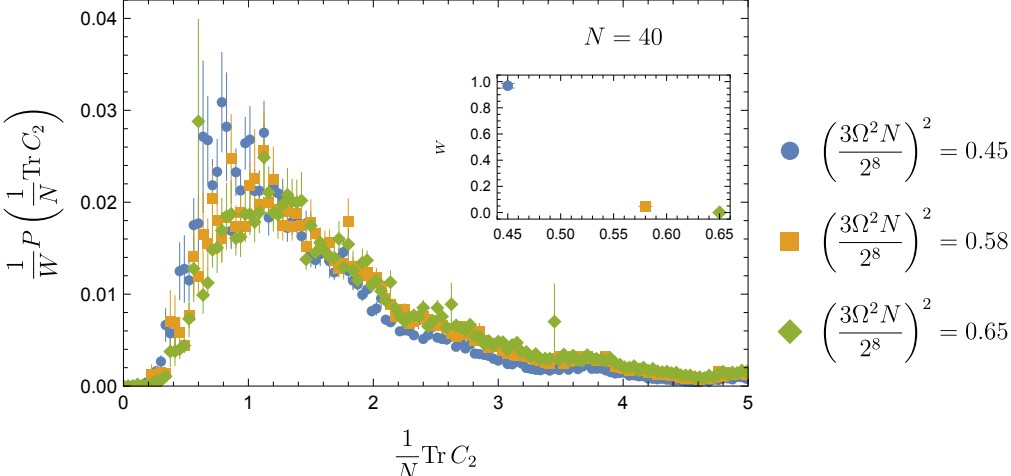

Figure 6: **Probability distribution of the Casimir** for values of $\Omega^2 N$ close to the phase transition, calculated using the full partition function at $N = 40$. The vertical lines are statistical error bars. The probabilities have been divided by the low Casimir weight $W$, which is the sum of the probabilities for all values of the Casimir up to 5 (the upper end of the plot). The inset shows the collapse of W as the transition is crossed.

in the low $\Omega$ phase as the maximally reducible or trivial representation, we have just seen that in fact it is more accurately described as almost-trivial or low-lying.

## 4 The Casimir distribution at intermediate deformation

As $\Omega^2 N$ is increased beyond the small values considered in the previous section, the approximation (19) breaks down. In Fig. 6 we see, however, that the distribution among low Casimir representations retains a qualitatively similar form to the small $\Omega^2 N$ distribution in the previous Figs. 4 and 5, now peaked at $\frac{1}{N} \text{Tr}\, C_2 = \mathcal{O}(1)$. The dominant representation can be identified by considering the occupation numbers $\langle n_M \rangle$. At $(\frac{3}{2^8} \Omega^2 N)^2 = 0.45$ and $N = 40$, as in Fig. 6, we find $\langle n_1 \rangle = 17.3$, $\langle n_2 \rangle = 5.1$, $\langle n_3 \rangle = 2.0$, and so on. As in §3, the one-dimensional representation is the dominant building block, with a small number of low-lying irreducible representations above it. The more dramatic effect visible in Fig. 6 is that as the phase transition is crossed, the total weight $W$ in representations with low Casimir drops to almost zero. This occurs because the maximal irreducible representation with $\frac{1}{N} \text{Tr}_{\mathcal{R}} C_2 = \frac{1}{4} N^2$, not shown in the plot, starts to become populated. This is the first order phase transition.

Before elaborating on the spacetime interpretation of these low Casimir representations, we briefly explain why the grand canonical ensemble — which we employed effectively in §3 — is not useful to capture the high Casimir representations that dominate above the phase transition. To describe these representations one must include the on-shell action (2), which becomes large. Naïvely proceeding as in §3, one finds to leading exponential order that

$$\langle n_M \rangle \approx e^{-\beta M + \frac{9\Omega^4}{2^{15}} M^3} . \tag{31}$$

The Boltzmann suppression in (31) is not enough to prevent unbounded growth of $\langle n_M \rangle$ at large $M$. The unconstrained sum in (25) is therefore divergent for all $\beta$. This super-Hagedorn growth is why the maximally irreducible representation so immediately becomes dominant above the transition in Figs. 2 and 3. The grand canonical partition function can be salvaged

by imposing $M \leq N$. Such a cutoff is consistent with the fundamental constraint that the representations must partition $N$, and leads to $\beta \approx \frac{9\Omega^4}{2^{15}}N^2$. However, this cutoff does not resolve the problem because the dominance of configurations with $M \sim N$ then produces a variance in $N$ equal to the expectation value squared: $\sigma_N^2 \equiv \frac{\partial^2}{\partial \beta^2}\log \widetilde{Z} \approx \sum_M M^2 \langle n_M \rangle \approx N^2$. The grand canonical and canonical ensembles are therefore not equivalent in this regime.

# 5 Representations and spacetime charges

We have started from the partition function (1), written in terms of $\mathfrak{su}(2)$ representations. However, this structure is not initially manifest in the matrix integral itself, but arises upon localising to supersymmetric saddles [3]. While at asymptotically large $\Omega$ each irreducible representation has an associated semiclassical matrix 'fuzzy sphere' configuration [1], at general $\Omega$ the fluctuations about these configurations are large. In particular, away from the large $\Omega$ limit the expectation value of $\frac{1}{N}\text{Tr}\,C_2$ is not equal to the matrix expectation value $\sum_{a=8}^{10}\frac{1}{N}\text{Tr}X_a^2$ (this is in units where the matrix action has an overall factor of $\Omega^4$). In the small $\Omega$ limit, for example, we have seen that the Casimir goes like $\Omega^2 N$ while the matrix expectation value goes like $1/\Omega^4$ [3]. Our above plots of the Casimir distribution will not then, in general, be equivalent to plots of the matrix expectation value such as those shown in [8]. As we now recall, the physics of the $\mathfrak{su}(2)$ representation is instead to specify how the $N$ units of D-instanton charge are carried in spacetime. The phase transition we have found is then understood as a change in nature of the spacetime charge carriers.

Every representation $\mathcal{R}$ has a corresponding spacetime geometry with nontrivial cycles that support different fluxes [2]. This connection works similarly to the LLM and LM geometries [21,22] — the representation determines the charge distribution for a Poisson problem that in turn determines the supergravity fields. For each value of $M$ appearing in the representation there is a seven-cycle carrying

$$N_{\text{D1},M} = n_M, \tag{32}$$

units of D1 flux. Let $M_1 < M_2 < \cdots < M_t$ be the values of $M$ that appear in the representation. Each sequential pair of these has a corresponding three-cycle carrying

$$N_{\text{NS5},s} = M_s - M_{s-1}, \tag{33}$$

units of NS5 flux. There is, furthermore, D5 flux that is fixed in terms of the D1 and NS5 flux. There is no F1 flux. Equations (32) and (33) show that the partition $N = \sum_M n_M M$ can, simultaneously, be interpreted as the D-instanton charge being partitioned into either D1- or NS5-branes. We will now elaborate on this fact.

The D1- and NS5-branes can be thought of as emergent from the large $N$ matrices, which initially describe $N$ D-instantons. For each $M$, the supersymmetric saddle point matrix configuration involves $n_M$ copies of a fuzzy sphere. At large $N$ the fuzzy spheres become the worldvolume of $N_{\text{D1},M}$ coincident spherical D1-branes, giving (32). For the same fixed $M$, the localisation formula (1) reduces the fluctuations about the fuzzy sphere saddle to $n_M$ moduli integrals over $m_{Mi}$. These integrals can be written in terms of a collective field

$$\rho_M(x) \equiv \sum_{i=1}^{n_M} \delta(x - m_{Mi}). \tag{34}$$

The geometry of the NS5-branes in (33) is encoded in these collective fields, as we discuss below and as was established for the BMN model in [23]. The fact that the NS5-branes emerge as fluctuations of the D1-branes highlights a kind of complementarity at work. The most convenient description depends on the representation.

Table 1: Parameter regime over which the spacetimes corresponding to the two extremal representations have small dilaton and are weakly curved close to the non-trivial cycles. Here Ric is the Ricci scalar in the string frame. All factors of $g_s$ are incorporated into $e^\phi$.

| Representation | $e^\phi \ll 1$ | Ric $l_s^2 \ll 1$ |
|---|---|---|
| Max irreducible | $N^{1/2} \ll \Omega^2 N$ | $\Omega^2 N \ll N^{1/2}$ |
| Max reducible (trivial) | $N^{1/4} \ll \Omega^2 N$ | $\Omega^2 N \ll N^{3/2}$ |

The limiting cases that control the phase transition in Fig. 2 are especially simple. The maximally irreducible representation has $N_{\text{D1}} = 1$ and $N_{\text{NS5}} = N$, while the trivial representation has $N_{\text{D1}} = N$ and $N_{\text{NS5}} = 1$. We can ask whether the corresponding geometries are good supergravity backgrounds over the parameter regimes where the representation in question dominates the partition function. The behaviour of the curvature and dilaton in the background for general representation is discussed in [2]. We collect the relevant results in Table 1. It is seen that the spacetime for the maximal irreducible representation is never a good supergravity background, in the sense that it is either stringy or quantum. The D1-brane backreacts gravitationally before the transition, at $\Omega^2 N \sim N^{1/2}$, but does not enter a classical gravitational regime. In Fig. 1 we have denoted the entire regime below the classical probe D1-brane description as a 'fluctuating D1-brane'. The first to onset are worldvolume stringy fluctuations [1] and then later bulk gravitational fluctuations. On the other hand, the spacetime for the trivial representation is a good background over the range $N^{1/4} \ll \Omega^2 N \ll N^{3/2}$ (in the following §6 this range of validity will be reduced by a further consideration). However, this range is strictly above the phase transition at $\Omega^2 N \sim 1$, and hence the trivial representation is sub-dominant there.

The same range of validity, $N^{1/4} \ll \Omega^2 N \ll N^{3/2}$, was obtained from a matrix perspective in [3]. There it arises as the condition for the collective field (34) to have a classical saddle point in the trivial representation and, more generally, low-lying representations with order one charge $N_{\text{NS5}} = q$. Specifically, it is found in Appendix D of [3] that

$$\rho_q(x) \sim (\hat{R}^2 - x^2)^2 \,. \tag{35}$$

We will discuss the value of $\hat{R}$ in the following §6. It determines the size of the conducting plates in the supergravity Poisson problem in the limit $z_s \ll R_s$, in the notation of [3]. However, the form of (35) suggests an additional interpretation. The collective field is the eigenvalue distribution of a matrix that carries an $SO(7)$ label. The distribution (35) is therefore naturally thought of as a uniform six-sphere embedded in $\mathbb{R}^7$ and projected down to one of the Cartesian axes. In §6 we will show, by working in a dual $SL(2,\mathbb{Z})$ frame, that $\hat{R}$ is indeed precisely the radius of a spherical $(p,q)$ fivebrane.

## 6   The fivebrane saddle

We have just seen that the maximally irreducible representation is not described by a good supergravity background at any value of $\Omega$. However, at large $\Omega^2 N \gg N^{3/2}$ it can be described by a probe D1-brane in a simple 'cavity' background, carrying $N$ units of worldvolume flux [1]. In this section we obtain an analogous brane description of the trivial and low-lying representations, by considering a $(p,q)$ fivebrane in the same cavity background. This brane configuration plays a similar role to the fivebrane in the BMN model [24]. We will first present

the solution and afterwards discuss its regime of validity in relation to the supergravity solutions.[1]

The cavity background [1] has a constant NSNS three-form field strength

$$H_3 = \mu\, dx^8 \wedge dx^9 \wedge dx^{10}, \tag{36}$$

where $\mu$ is a parameter that is proportional to $\Omega$ in the matrix description. The dilaton, axion and string frame metric are given by

$$e^{\phi} = -\frac{1}{C_0} = g_s\left[1 - \frac{\mu^2}{32}\left(\sum_{A=1}^{7} x_A^2 + 3\sum_{a=8}^{10} x_a^2\right)\right], \qquad ds^2_{str} = \sum_{i=1}^{10} \frac{e^{\frac{1}{2}\phi}}{\sqrt{g_s}} dx_i^2. \tag{37}$$

This background is very similar to the IIA geometry obtained by dimensional reduction of the BMN plane wave [28]. We are using a convention where the factors of $g_s$ are part of the background fields and do not appear in the action explicity. This will make the duality transformations we perform shortly more transparent.

A probe D1-brane can be added to the cavity background, preserving all symmetries and supersymmetries. The brane forms an $S^2$ with radius $r^2 \equiv \sum_{a=8}^{10} x_a^2$, sitting at the origin in the other directions, and carries a worldvolume Maxwell flux $\mathcal{F}$ given by [1]

$$r = \tfrac{3}{4}\pi\alpha'\mu N, \qquad \mathcal{F} = \tfrac{1}{2}N\text{vol}_{S^2}. \tag{38}$$

The D1-brane sources $N$ units of D-instanton charge, via the $\int C_0 \mathcal{F}$ worldvolume coupling, and $N_{\text{D1}} = 1$ unit of electric RR flux through the $S^7$ transverse to the brane, via the $\int C_2$ worldvolume coupling. It therefore has both the charge and the geometry of the maximal irreducible representation. The D5 and NS5 charges in this case can only be seen after backreacting the brane on the spacetime.

To obtain a brane description of other representations, we will now place a $(p,q)$ fivebrane in the cavity background. Such a configuration has $N_{\text{NS5}} = q$. While our main interest is in the trivial representation, with $q = 1$, it is instructive to consider this more general case. The D5 charge $p$ is unfixed at this point. To write down the brane action, following the logic of Polchinski and Strassler [29], we will perform the $SL(2, \mathbb{Z})$ transformation (see [1] for more on $SL(2, \mathbb{Z})$ in Euclidean type IIB), where $F_3$ is the RR three-form field strength,

$$\widetilde{C}_0 \pm e^{-\widetilde{\phi}} = \frac{a(C_0 \pm e^{-\phi}) + b}{-q(C_0 \pm e^{-\phi}) + p}, \qquad \begin{pmatrix} \widetilde{H}_3 \\ \widetilde{F}_3 \end{pmatrix} = \begin{pmatrix} p & -q \\ b & a \end{pmatrix}\begin{pmatrix} H_3 \\ F_3 \end{pmatrix}, \tag{39}$$

with $ap + bq = 1$. Integer solutions for $a$ and $b$ exist whenever $p$ and $q$ are coprime, which is sufficient for our purposes. This transformation maps the $(p,q)$ fivebrane to a single probe D5-brane in a transformed cavity background. It is straightforward to obtain $\widetilde{H}_3, \widetilde{F}_3, \widetilde{C}_0$ and $e^{-\widetilde{\phi}}$ from (39) in terms of the original background (36) and (37). This original background has $F_3 = 0$. The $(p,q)$ fivebrane action is then the D5-brane action in the transformed background

$$S_{(p,q)} = \widetilde{S}_{\text{D5}} = T_5\left[\int d^6\sigma\, e^{-\widetilde{\phi}}\sqrt{\det\widetilde{\mathcal{G}}} + \int \widetilde{\mathcal{C}}_6\right]. \tag{40}$$

Here $T_5^{-1} = (2\pi)^5\alpha'^3$ [30], $\widetilde{\mathcal{G}}$ is the pull-back of the string frame metric and $\widetilde{\mathcal{C}}_6$ is the pull-back of $\widetilde{\mathcal{C}}_6$, which obeys $d\widetilde{\mathcal{C}}_6 = \star e^{\widetilde{\phi}}(\widetilde{F}_3 - \widetilde{C}_0\widetilde{H}_3)$. There are no explicit factors of $g_s$ in the action, as these have been incorporated into the background fields above.

---

[1]Further recent developments in supergravity for IKKT holography include [25]. Tunneling into polarised brane configurations has been discussed from a worldvolume perspective in works including [26,27].

We look for a brane embedding that preserves the $SO(3) \times SO(7)$ symmetry of the background by sitting at a fixed $R^2 \equiv \sum_{A=1}^{7} x_A^2$, and at the origin in the other directions. As in [1] we will furthermore look for embeddings away from the singular boundary of the cavity, with $R\mu \ll 1$. Substituting the transformed fields into the action and taking this limit we obtain

$$S_{(p,q)} \approx \frac{1}{g_s^2} \frac{R^6}{30\pi^2 \alpha'^3} \left[ \sqrt{g_s p \left( g_s p + 2q \right)} - \frac{R\mu}{7} g_s p \right]. \tag{41}$$

Here we used the fact that the volume of a six-sphere is $\frac{16\pi^3}{15}$.

To fix the radius $R$ in (41) we must impose that the $(p,q)$ fivebrane source $N$ units of D-instanton charge. The D-instanton charge is associated to an $SL(2,\mathbb{Z})$ $T$-transformation under which $C_0 \to C_0 + X$ and $F_3 \to F_3 + X H_3$. That is,

$$-T_{-1} N = \frac{\partial S_{(p,q)}}{\partial C_0} + \frac{\partial S_{(p,q)}}{\partial F_3} H_3. \tag{42}$$

Here $T_{-1} = 2\pi$ is the D-instanton 'tension'. The negative sign is because the cavity background as given above leads to $N$ anti-instantons, see footnote 5 in [1]. Using the action (40), with the cavity background transformed by (39), and then taking $R\mu \ll 1$ gives

$$N \approx \frac{1}{g_s} \frac{q R^6}{60\pi^3 \alpha'^3} \left[ \frac{g_s p + q}{\sqrt{g_s p \left( g_s p + 2q \right)}} - \frac{R\mu}{7} \right]. \tag{43}$$

Solutions to (43) with small $R\mu$ arise in the limit $g_s p \ll 1$, with $q$ fixed. In this limit we find

$$\frac{R}{l_s} \approx g_s^{1/4} \left( 5! \pi^3 \right)^{1/6} \left( \frac{p N^2}{2 q^3} \right)^{1/12}. \tag{44}$$

Here we introduced the string length $l_s \equiv \sqrt{\alpha'}$. It is interesting to note here that $g_s^{1/4} l_s = l_{\mathrm{Pl}}$, the Planck length. Plugging (44) into the action (41) gives

$$S_{(p,q)}^{\text{on-shell}} \approx 4\pi N \frac{p}{q}. \tag{45}$$

The value of $p$ will be fixed once the backreaction of the $(p,q)$ fivebrane on the geometry is considered. This is beyond what we wish to consider here. The backreacted geometries in [2] have $p = \frac{1}{512} \left[ 15^2 \Omega^{12} N^2 q^3 / (2\pi^3) \right]^{1/5}$. If we use this value in (44) the radius becomes

$$\frac{R}{l_s} \approx (2\pi g_s)^{1/4} \left( \frac{15\pi}{8} \frac{N\Omega}{q} \right)^{1/5}. \tag{46}$$

The radius $R$ in (46) exactly matches $\hat{R}$ in the collective field (35) upon converting from matrix to spacetime units [2]. On the other hand, the action (45) is $\frac{14}{5}$ times the action obtained from the collective field in [3]. We believe that this mismatch is due to the fact that we have not accounted for the dynamics that fixes $p$, which will contribute to the action.

We now discuss the regime of validity of the probe D5-brane. We will verify that the probe brane and supergravity descriptions are complementary, as one might expect. The supergravity solutions require a large D5-brane charge, so that $p \gg 1$ [2]. The expression for $p$ given above then gives a new constraint for validity of the supergravity solution, $N^{2/3} \ll \Omega^2 N$. This reduces the lower end of the supergravity validity window in Table 1. Now consider the D5-brane probe. In our limit, $g_s p \ll 1$, the dilaton in the transformed frame is

$$e^{\tilde{\phi}} \approx 2pq. \tag{47}$$

Remarkably, this expression is independent of $g_s$. For the D5-brane description to be weakly coupled, with an order one $q$, we therefore require[2] $p \ll 1$ and hence $\Omega^2 N \ll N^{2/3}$. The range of validity of the collective field saddle (35), above we recalled that this is $N^{1/4} \ll \Omega^2 N \ll N^{3/2}$, is therefore partitioned into a lower range where it describes a probe D5-brane and a higher range where it describes a supergravity geometry. This partition has been illustrated in Fig. 1. Both the D5-brane and the supergravity background match the radius of the collective field saddle.

# 7 Discussion

The polarised IKKT model exhibits a first order large $N$ phase transition at $\Omega^2 N \sim 1$. The transition is an exchange of dominance between the maximally irreducible and maximally reducible $\mathfrak{su}(2)$ representations. These different representations correspond to different partitions of the $N$ units of D-instanton charge into D1- and NS5-branes. While each of these partitions has a corresponding supergravity background, in no parameter regime is the dominant representation dual to a weakly curved background with small dilaton. Above the transition, the spacetime description of the matrices is a single D1-brane that is either non-backreacting or quantum fluctuating. Below the transition the spacetime description is a gas of D-instantons. The D-instantons in the trivial representation dominate because in the higher representations some of the instantons form bound states and therefore have a lower dimensional moduli space.

The holographic emergence of semiclassical spacetime is the process by which matrix degrees of freedom construct their own geometry rather than living in a pre-existing background. The previous paragraph suggests that in the polarised IKKT partition function such physics occurs only within sub-dominant matrix configurations. Perhaps this was to be expected: it has recently been emphasised that good semiclassical gravity backgrounds can have negative Euclidean modes [31], indicating that they also arise as a sub-dominant configuration in some microscopic theory. In the spirit of that paper, one way to access the sub-dominant geometries may be to ask relational questions of the theory. These would condition the partition function on the value of specific observables. A natural class of observables to consider are supersymmetric quantities that can be computed within the localisation framework that leads to the partition function (1). Such observables are discussed in [3]. A relational approach to the matrix partition function also has the potential to generate an emergent quantum mechanical Wheeler-DeWitt time.

A weaker notion of emergent geometry is realised by the probe branes that we have discussed above. Here the matrices do not manage to gravitate, but are able to create a semiclassical geometry that supports collective excitations, including worldvolume gauge fields. One may wonder whether the gapless worldvolume excitations, described in [1], are 'matrix' Goldstone bosons, associated to spontaneous symmetry breaking in the large $N$ matrix integral. In particular, the spherical probe D1-brane that dominates at large $\Omega$ has an excitation spectrum that goes to zero with the angular momentum quantum number $l$. The $l = 0$ mode is a rotational zero mode, which acts nontrivially on the fuzzy sphere matrix configuration. Recall that the fuzzy sphere is classical at large $\Omega$, with radius squared $\sum_{a=8}^{10} \frac{1}{N} \mathrm{Tr} X_a^2 \sim N^2$ (again, in units where the matrix action has an overall factor of $\Omega^4$). These two facts — a manifold of vacua and a tower of low-action excitations emanating from the zero modes —

---

[2]The constant $p$ need not be integer in the probe D5-brane description. Clearly, when $p \ll 1$ the probe D5-brane is no longer related to a $(p, q)$ fivebrane in the cavity by an $SL(2, \mathbb{Z})$ transformation. In this regime we should think of the probe D5-brane in the transformed cavity, which is a solution in its own right, as the appropriate spacetime description of the matrix saddle, with the correct geometry and charges.

both suggest spontaneous breaking of the $SO(3)$ symmetry of model. The story is further enriched by the additional $U(N)$ symmetry. This symmetry also acts nontrivially on the fuzzy sphere configurations. A $U(N)$ transformation can be used to undo the rotational zero mode, but acts as a combined Maxwell gauge transformation and area-preserving diffeomorphism on the low-lying excitations.

In contrast, any spontaneous symmetry breaking of the $SO(7)$ symmetry by the probe fivebranes is less manifest. The classical degrees of freedom in this case are not the matrix components but rather the collective field (34). While the collective field does pick out a direction in $SO(7)$, this is because the localising term in the action breaks the symmetry explicitly (by construction, this localising term does not alter the partition function). Given that the fivebranes also admit worldvolume fields, it would be good to understand the symmetry dynamics better in this case. The collective field furthermore describes the supergravity regime, as we have discussed in §6, and therefefore a matrix symmetry-breaking understanding of this regime may also shed light on the emergence of gravity.

We will end with a comment on the relationship between the cavity geometry, discussed in [1] and §6, and the bubbling geometries constructed in [2]. It was noted in [2] that the cavity geometry can be obtained by dualising the asymptotic expansion of the bubbling geometries. However, in §6 the radius of the $(p, q)$ fivebrane in the cavity matched — as a function of the D1, NS5 and D5 charges — with a radius in the bubbling geometries for low-lying representations. This fact suggests that the cavity and the bubbling geometry descriptions should be reconciled within the same $SL(2, \mathbb{Z})$ duality frame. The previous match in [1] of a probe D1-brane in the cavity with the maximally irreducible representation also suggests that the bubbling geometries and the cavity are in the same duality frame. How this works remains to be fully understood.

# Acknowledgments

It is a pleasure to acknowledge discussions with Shota Komatsu, Adrien Martina, Joao Penedones, Antoine Vuignier and Xiang Zhao about their results.

**Funding information**   This work has been partially supported by STFC consolidated grant ST/T000694/1. SAH is partially supported by Simons Investigator award #620869. JL is supported by a Harding Distinguished Postgraduate Scholarship.

# A   Finite size scaling towards the critical point

The estimate (17) for the critical value of the deformation parameter at infinite $N$ was obtained as follows. For several values of $N$ the curve for the Casimir was fitted by a tanh function close to the transition. Some illustrative fits are shown in Fig. 7, the important point is to capture the crossover region. The centre of the tanh functions were then fitted to a quadratic function of $1/N$. This fit is shown in Fig. 8. The $N = \infty$ value of 0.58, quoted in (17), is obtain as the intersection of the fit with the $y$ axis. Note that both a $1/N$ and a $1/N^2$ term are needed to obtain a decent fit.



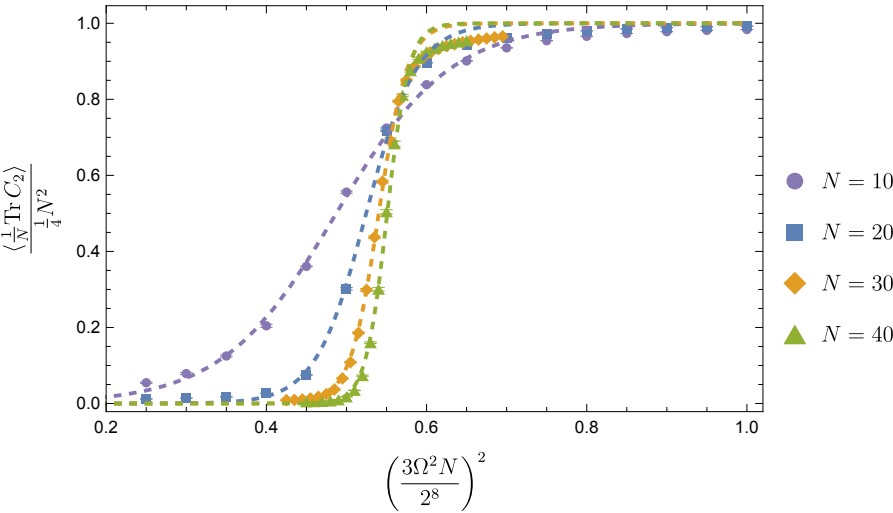

Figure 7: **Near-transition fits** of the Casimir to a tanh function, for several values of $N$.

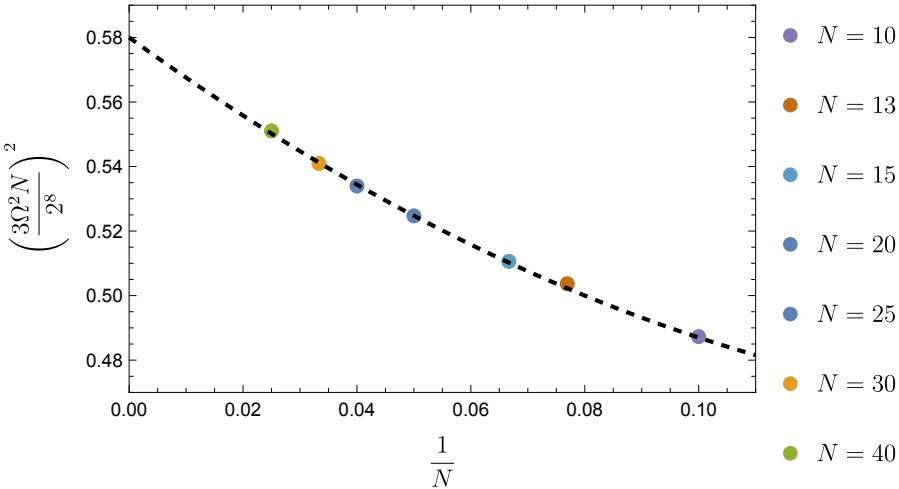

Figure 8: **Centre of the transition as a function of** $1/N$, extracted from the tanh fits. The dashed curve fits the data points shown to $y = 0.58 - 1.3x + 3.75x^2$.

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
