# Peer review of "Statistical Physics of the Polarised IKKT Matrix Model"

_SciPost Physics, doi:SciPost Phys. 19, 099 (2025)_

## Round 2 · Referee Report · Anonymous (Referee 1) · 2025-7-6

Report
The IKKT matrix model was proposed in 1997 as a nonperturbative formulation of superstring theory, and as such, it has been investigated by many people both analytically and numerically. I suggest the authors to quote some important papers in this direction since it would help the readers understand in what context the IKKT model has been investigated so far.
The polarized IKKT model is a model that one can obtain by deforming the original IKKT matrix model in such a way that 16 supersymmetries are maintained. The recent excitement about this SUSY deformed model is that the dual geometry corresponding to each saddle point of the matrix model has been identified. Unlike the conventional duality between gauge theory and gravity theory pioneered by Maldacena, the matrices in the IKKT model do not depend on time. Hence there is neither space nor time on the gauge theory side. It is this aspect that makes the new duality more interesting. However, the duality discussed so far is purely in the Euclidean setup. It would be even more exciting if this duality is extended to the Lorentzian setup since one may then investigate the emergence of not only space but also time.
While this goes beyond the scope of the present paper, it would be better to be mentioned at least as an interesting future prospect.
Another important aspect of the polarized IKKT model is that one can obtain the partition function explicitly by using the localization technique making use of the supersymmetries that are preserved in the deformation. Thus the partition function reduces to the integration over the moduli parameters around each saddle, which can be evaluated by Monte Carlo sampling much more easily than simulating the original matrix model. Using such a method, the authors discover a phase transition at some critical Omega, where Omega is the deformation parameter, which is most likely of first order. This may be supported by the standard finite size scaling, which in the present case amounts to confirming the shift of the critical point with O(1/N^2). Since the authors have results for some values of N, they may try to see if the shift is consistent with this scaling.
At larger Omega, the maximal fuzzy sphere configuration dominates since it gives the minimal action. One should note that by rescaling the matrices as A->Omega A, Psi->Omega Psi, one can factor out the Omega dependence of the action as Omega^4, so that at large Omega, the classical solutions should dominate. This is not mentioned in this paper, however. At small Omega, the almost-trivial saddles dominate. As they emphasize in this paper,
this statement should be taken with care since it does not refer to the dominant matrix configurations in the original matrix integral at small Omega. The authors discuss this point carefully by considering the grand canonical ensemble, which is tractable at small Omega.
Finally the authors discuss the gravity dual picture in detail and find that the supergravity picture is only valid in some region of Omega for the subdominant saddle point. This is a bit unfortunate, but it is not something that diminishes the value of this paper.
I have some more suggestions for improvements. First I strongly recommend them to write the IKKT matrix model and its supersymmetric deformation in the Introduction to make the paper self-contained. This will make it easier to discuss some past work on the model.
Second I noticed some typos. I think the authors should use a spell-checker to avoid problems like "non-backreating" mentioned below.
p.3, below eq.(4)
The constant ....
This is not a full sentence.
p.4, caption of Fig.1
"The various regimes shown are discussed throughout the paper, this figure may be useful as a roadmap."
I think "and" is needed after comma.
p.17, section 7
non-backreating -> non-backreacting
Recommendation
Ask for minor revision

---

## Round 2 · Referee Report · Anonymous (Referee 2) · 2025-7-15

Report
The analyses in the paper provide new insights into the gauge/gravity duality for the (polarised) IKKT model and will affect our general understanding of emergent spacetime. Therefore, I think the article is suitable for publication.
Recommendation
Publish (meets expectations and criteria for this Journal)

---

## Round 3 · Author Response

We thank both of the referees for their positive reports.

Referee #2 has recommended publication in its current form.

Referee #1 has asked for minor revisions. We have addressed these issues, or not, as follows.

  1. We have corrected the two typos pointed out by the referee. We have, however, retained the sentence “The constant c_N = … .” below equation (4). Our understanding is that within a mathematical context it is acceptable to use the = sign of an equation as the verb in a sentence.

  2. The referee asks us to perform finite size scaling to validate the first order transition, by plotting the dependence of the critical point on 1/N^2. This is what we had done, and we have now included the analysis leading to equation (17) in a new short appendix A. It can be noted that there are 1/N corrections, not just 1/N^2.

  3. The referee emphasises that the IKKT model has been worked on by many people and “suggest the authors to quote some important papers in this direction since it would help the readers understand in what context the IKKT model has been investigated so far.” Furthermore the referee “strongly recommend them to write the IKKT matrix model and its supersymmetric deformation in the Introduction to make the paper self-contained. This will make it easier to discuss some past work on the model.” And finally they suggest “It would be even more exciting if this duality is extended to the Lorentzian setup since one may then investigate the emergence of not only space but also time. While this goes beyond the scope of the present paper, it would be better to be mentioned at least as an interesting future prospect.”

This our second paper on the polarised IKKT model. In our first paper (arXiv 2409.18706) we indeed wrote down the model in great detail, including careful discussion of spinor representations, supersymmetry etc. In that paper we also included an extended list of references of what we believed to be the most interesting of the many previous papers on the model, understood generally. That paper also included an explicit section on the Lorentzian model.

In the present paper we have limited ourselves, for the most part, to references that are directly relevant to what we are doing. Within that scope, we have cited generously. Readers interested in a broader context can look at our previous paper for entry points to the extensive literature.

The starting point of the present paper is the partition function in equation (1) — all quantities in this equation are defined explicitly in the paper and in this sense our paper is already self-contained. All papers have a starting point and do not need to be ab initio.

In our opinion, adding more background or tangential material to the introduction of the paper would have the effect of obscuring what we are doing. So, respectfully, we have not made any changes related to these points of the referee.

  1. The referee makes the following comment: “Finally the authors discuss the gravity dual picture in detail and find that the supergravity picture is only valid in some region of Omega for the subdominant saddle point. This is a bit unfortunate, but it is not something that diminishes the value of this paper.”

We do not agree that this is unfortunate. It seems quite likely that we are not living in a dominant saddle of whatever the theory of our true universe is, so this fact may well be a plus. As we discuss in the paper, we think that the work that needs to be done relating to this point is to build relational observables to access sub-dominant saddles and, at the same time, describe an emergent time.

---

## Round 3 · List of Changes

See above.

---

## Editorial Decision

published